# A One-Year Retrospective Analysis of Viral and Parasitological Agents in Wildlife Animals Admitted to a First Aid Hospital

**DOI:** 10.3390/ani13050931

**Published:** 2023-03-04

**Authors:** Maria Irene Pacini, Maurizio Mazzei, Micaela Sgorbini, Rossella D’Alfonso, Roberto Amerigo Papini

**Affiliations:** 1Department of Veterinary Sciences, University of Pisa, Viale delle Piagge 2, 56124 Pisa, Italy; 2Department of Systems Medicine, University of Rome Tor Vergata, 00133 Rome, Italy

**Keywords:** wildlife, central Italy, viruses, helminths, protozoa, interspecies transmission, zoonotic pathogen

## Abstract

**Simple Summary:**

In recent decades, wildlife populations in Italy have continued to expand, and some species are now present in large numbers with a wide geographical distribution. Viral and parasitic agents are an integral part of any wildlife population. The major changes in human land use, the tendency of some wild animals to get closer to urban areas in search of food, the increased interest in visiting protected natural areas, and the hunting of game as a food source increase the possibility of sharing natural areas between wild animals, livestock, pets, and humans. From an epidemiological point of view, these factors also increase the possibility of the exchange of pathogens between these groups. Therefore, wild animals can act as a source of infection for domestic animals and humans. This study represents a retrospective analysis including viral agents and parasites affecting a cohort of wild animals in Italy, providing a comprehensive overview of their health status. Overall, a large number of animals tested positive for at least one pathogen, and many were infected with more than two agents, showing a wide range of pathogens responsible for intra- and interspecific transmission in wild populations living in the study areas.

**Abstract:**

This study aimed to provide information on the presence and frequency of viral and parasitic agents in wildlife presented to a Veterinary Teaching Hospital in 2020–2021. Serum and faecal samples were collected from 50 rescued animals (roe deer, fallow deer, foxes, badgers, pine martens, and porcupines) and examined by serological, molecular, and parasitological techniques. Transtracheal wash (TTW) was also collected post-mortem from roe deer. Overall, the results of the different techniques showed infections with the following viral and parasitic agents: Bovine Viral Diarrhea Virus, Small Ruminant Lentiviruses, *Kobuvirus*, *Astrovirus*, *Canine Adenovirus 1*, *Bopivirus*, gastrointestinal strongyles, *Capillaria*, *Ancylostomatidae*, *Toxocara canis*, *Trichuris vulpis*, *Hymenolepis*, *Strongyloides*, *Eimeria*, *Isospora*, *Dictyocaulus*, *Angiostrongylus vasorum*, *Crenosoma*, *Dirofilaria immitis*, *Neospora caninum*, *Giardia duodenalis*, and *Cryptosporidium*. Sequencing (Tpi locus) identified *G. duodenalis* sub-assemblages AI and BIV in one roe deer and one porcupine, respectively. Adult lungworms collected from the TTW were identified as *Dictyocaulus capreolus* (COX1 gene). This is the first molecular identification of *G. duodenalis* sub-assemblage AI and *D. capreolus* in roe deer in Italy. These results show a wide presence of pathogens in wild populations and provide an overview of environmental health surveillance.

## 1. Introduction

In recent decades the incidence of introduced and emerging infectious diseases in humans and animals has increased worldwide [1,2]. Wildlife plays an important role in the epidemiology of infectious and parasitic diseases, resulting in a source of various pathogens [1,3,4] with significant implications on human health, wild and domestic animal health, biodiversity, and economy, as it harbours most of the emerging and re-emerging pathogens and could play a significant role in their spread and persistence in ecosystems [1,5,6,7].

Numerous studies on wildlife have shown that wild animals act as hosts for some known and unknown pathogens that are transmissible to several other species, including domestic animals and humans. Therefore, in recent years, research on wildlife diseases has become an increasing international interest as a crucial aspect of wildlife conservation projects, such as reintroduction and translocation programmes, especially for species of high conservation value, and programs of disease surveillance for domestic animals and humans [1,8,9]. Unfortunately, the elusive and unique nature of these animals makes it difficult to sample and conduct health surveillance actions in these species and to understand their real role in the epidemiology and ecology of diseases [10].

Wildlife rescue centres can play an important sentinel role and represent an underutilised source of information on pathogens circulating in ecosystems, particularly at the wildlife–domestic animal interface [11,12,13]. Due to the increasing proximity of wild animals to urban areas, often in search of food, and the growing public awareness of biodiversity conservation, the number of wild animals admitted to rescue centres is steadily increasing. Each year, rescue centres recover hundreds of animals belonging to many different species, some of which were impossible to sample in the wild. In addition, wildlife rescue centres represent a system in which many injured, diseased, and stressed animals interact and where the risk of introducing, amplifying, and spreading pathogens is considerable. Similarly, the release of rescued animals into the wild habitats can lead to the spread of pathogens in the area, posing a health risk to the local population. It is, therefore, important to gather information on the pathogens that are or could be circulating among hospitalised animals.

Bovine viral diarrhoea virus (BVDV), family Flaviviridae, infects a variety of ungulate species and is associated with severe economic losses in livestock production worldwide. Several wild and domestic ruminant species cohabit during the grazing period, and BVDV circulation has been demonstrated in both domestic and wild ruminants and interspecific transmission has already been reported [14,15,16]. In Italy, a serosurvey was carried out in four large alpine ungulates in the High Valley of Susa, and antibodies were detected in chamois, wild boar, and red deer, but all roe deer were negative [15]. A similar result was obtained a few years later by Fernández-Sirera and colleagues.

Small Ruminant Lentivirus (SRLV), genus Lentivirus, family Retroviridae, are widespread throughout the world and have a detrimental economic impact on the small ruminant industry due to increased mortality and reduced animal performance [17,18].

*Bopivirus* is a novel genus of the family Picornaviridae recently discovered in faecal samples from wild and domestic ruminants. To date, genus *Bopivirus* contains the species *Bopivirus A*, detected in cattle, the novel *Bopivirus B*, detected in sheep and goats in Hungary; and the newest, tentatively designed *Bopivirus C*, detected for the first time in faeces of Australian fallow and red deer [19,20].

*Astroviruses* (AstV) are small, round, non-enveloped RNA viruses. First discovered in children with diarrhoea, they have been discovered in a variety of domestic and wild animals, both with and without gastroenteritis [21,22,23,24,25,26,27,28,29,30,31]. The wide host range and genetic diversity within the *Astroviridae* family have made classification difficult, and they are currently divided into two genera, *Mamastrovirus* (with 19 species) and *Avastrovirus* (with 3 species), infecting mammalian and avian hosts, respectively. However, numerous unclassified AstVs have yet to be accepted as species [32].

*Kobuvirus* is a genus belonging to the Picornaviridae family and consists of three species, *Aichivirus A* (formerly *Aichi virus*), *Aichivirus B* (formerly *Bovine kobuvirus*) and *Aichivirus C* (*porcine kobuvirus*). First recognised in humans with gastroenteritis, *kobuviruses* have recently been detected in a variety of animals in several countries with an ever-increasing number of hosts. In particular, they have been found in domestic and wild carnivores, ruminants, and suids [33,34,35,36,37,38,39,40,41,42].

*Canine adenovirus 1* (CAdV-1), family Adenoviridae, genus Mastadenovirus, is the causative agent of infectious canine hepatitis in dogs and is characterised by high pathogenicity due to differential tissue tropism. In addition to the dog, CAdV-1 infection has been documented in several free-ranging wild mammalian carnivores, foxes, wolves, brown bears, striped skunks, and Eurasian otters, causing from subclinical to fatal diseases and sporadic epizootics [43,44,45,46,47].

As far as parasitic agents are concerned, coccidia are protozoa with a direct life cycle that infect the epithelial cells of the intestinal tract of their hosts and that usually have high host-specificity. The following *Eimeria* species are found in roe deer: *Eimeria* capreoli; *Eimeria* catubrina; *Eimeria* panda; *Eimeria* patavina; *Eimeria* ponderosa; *Eimeria* rotunda; and *Eimeria* superba [48]. Instead, *Isospora canis*, *Isospora canivelocis*, *Isospora ohioensis*, *Isospora pavlodoratica*, *Isospora vulpina*, and *Isospora vulpis* may occur in red foxes, though it must be considered that coccidia of canids is an exceptionally confused group of coccidia [49]. *Eimeria melis* and *Isospora melis* have been reported in badgers [50].

Gastrointestinal strongyles (GSI) are very common among wild ruminants and always occur in mixed infections with different nematode species whose eggs are morphologically indistinguishable from one another. Infections with helminths belonging to the genera *Bunostomum*, *Chabertia*, *Haemonchus*, *Nematodirus*, *Oesophagostomum*, *Ostertagia*, and *Thychostrongylus* have been reported in the cervids [51]. Hosts get infected by direct ingestion of third-stage larvae whilst grazing. *Capillaria bovis* and *Capillaria bilobata* have been reported in roe deer [52,53]. This intestinal nematode of ruminants has a direct life cycle and is considered to be of low pathogenic significance.

*Dictyocaulus capreoli* has previously been reported as a parasitic agent of bronchopulmonary infection in roe deer [54,55]. In particular, this lungworm is a specific parasite for C. capreolus, as the name suggests, and *Alces* alces (moose) [54]. The life cycle of the *Dictyocaulus* species is direct. Adult lungworms live in the bronchi and bronchioles of their hosts. Females are ovoviviparous. First-stage larvae hatch in the respiratory or intestinal tract and are expulsed with the faeces into the environment. Hosts get infected by ingesting L3 larvae whilst grazing.

Intestinal worms (*Toxocara canis*, *Trichuris vulpis*, *Ancylostomatidae*), lungworms (*Capillaria* spp.), and heartworms (*Dirofilaria immitis*, *Angiostrongylus vasorum*) can be found in red foxes. Their life cycle can be direct (*T. vulpis*, *Capillaria* spp.), indirect (*D. immitis*, *A. vasorum*), or both (*T. canis*, *Ancylostomatidae*), depending on the species. Within the family *Ancylostomatidae*, both *Ancyclostoma caninum* and *Uncinaria stenocephala* have been reported in V. vulpes [56].

Among *Capillaria* species, both *Capillaria aerophila* (syn. Eucoleus aerophilus) and *Capillaria boehmi* (syn. *Eucoleus boehmi*) are commonly found in the respiratory system of fox populations [57,58].

Overall, within the family *Ancylostomatidae*, four species belonging to the *Uncinaria* or *Tetragomphius* genera [59] and unidentified species belonging to the Ancylostoma genus [60] have been reported in badgers. *Uncinaria criniformis* is the species widely reported in European badger populations [59,61,62,63].

Lungworm species reported in Meles include *Crenosoma melesi* [63] and *Crenosoma vulpis* [64], as well as C. aerophile [60]. *Crenosoma* spp. has an indirect life cycle with terrestrial gastropods as intermediate hosts, while C. aerophila has a direct life cycle.

*Strongyloides* is a genus of small parasitic nematodes infecting the small intestine of mammals, characterised by an unusual life cycle involving one or more generations of free-living adult nematodes. *Strongyloides* sp. infections have been reported in European badgers [61,62]. Results of phylogenetic analysis in different carnivorous hosts suggested that badgers can be infected with a distinct species of *Strongyloides* [65].

The European badger may also be a definitive host for three species of heartworms. These include *D. immitis* as well as *A. vasorum* and *Angiostrongylus daskalovi*, which was considered a relatively rare species [61,62,66,67,68]. *Angiostrongylus* spp. has snails and slugs as intermediate hosts, while *D. immitis* is transmitted by mosquito bites. Reported prevalence values in badgers are 0.87% for *D. immitis*, 6.4 to 24% for *A. vasorum*, and 72% for *A. daskalovi* [61,62,66,67].

*Neospora caninum*, an apicomplexan protozoan, is among the main causes of abortion in cattle and has both horizontal and vertical transmission patterns. The existence of a sylvatic life cycle of N. caninum has been supported by some authors [69].

*Giardia duodenalis* is a flagellate protozoan with a direct orofecal life cycle. Molecular investigations have revealed that *G. duodenalis* is a complex species, including eight morphologically indistinguishable genot ypes or assemblages with different host ranges classified as A to H. Genotypes A and B infect humans but can also be found in animals with their respective sub-assemblages [70,71].

The Veterinary Teaching Hospital (VTH) of the Department of Veterinary Sciences of the University of Pisa has been providing a 24-h emergency service for wild mammals rescued in the Pisa area since 2010. The aim of the present study was to retrospectively investigate the presence and frequency of different virological and parasitic agents in a cohort of 50 injured wild mammals admitted to the VTH from September 2020 to August 2021. In addition, the study area was divided into three zones based on the different levels of urbanisation, population density, and land use and differences in the prevalence of viral pathogens concerning the rescue area were also investigated.

## 2. Materials and Methods

### 2.1. Study Area

Tuscany is the region of the Italian peninsula with the highest number of wild ungulates, with about 400,000 animals in constant increase, and it is estimated that 40% of roe deer, 45% of fallow deer, and 30% of wild boar of the entire country live there [72]. The province of Pisa, composed of 39 municipalities, covers an area of 2450 km^2^; 25% of the territory is made up of plains, with most of the inhabited centres, and the rest of the territory consists of hills and mountains suitable for the development of numerous and diverse wildlife populations [73]. Despite this, the data on land use shows that almost the entire territory of the province is used for industrial or agricultural purposes and that unused part is very limited [73]. Therefore, the uncontaminated and uninhabited areas are very limited, and wildlife habitats are fragmented and interrupted by an extensive network of roads and urban areas with many opportunities for contact and interaction between wildlife, domestic animals, and humans. Moreover, the territory of the province of Pisa is very diverse due to its considerable size, with the main urban settlements located in the northern area, where, in addition to the city of Pisa, some of the most important inhabited centres are located. For this study, the province was divided into three zones of similar size, characterised by different demographic densities, road networks, and land use (Figure 1, Figure 2, Figure 3 and Figure 4).

The zones have been named Zone A, B, and C (Figure 1). Data on zone extension, number of inhabitants, population density, and forest cover, expressed as its relationship with the total area surface, are shown in Table 1 [73,74].

### 2.2. Enrolled Animals

In this study, a cohort of 50 wild animals, injured and hospitalised during a one-year period (between September 2020 and August 2021), was retrospectively assessed for the presence and frequency of different virological and parasitic agents. The enrolled animals were older than 12 months (based on general conformation and teeth eruption), were of both sexes, and belonged to the following species: 23 roe deer (*Capreolus capreolus*); 4 fallow deer (*Dama dama*); 12 red foxes (*Vulpes vulpes*); 6 badgers (*Meles meles*); 4 porcupines (*Hystrix cristata*); and 1 pine marten (*Martes martes*). For each animal, information on the place and the cause of the rescue, as well as on age, sex, and species, was collected from the person who brought the animals (volunteers, citizens, local authorities) and during the first clinical examination and recorded in the hospital database.

Data on the species and sex of the enrolled cohort are presented in Table 2.

All the animals enrolled were hospitalised because they were injured. Overall, 34/50 (68%) of animals were injured in car accidents, 3/50 (6%) by predators, and 2/50 (4%) by fighting with conspecifics. In 11/50 (22%) animals, the cause for injuries could not be readily determined.

### 2.3. Sample Collection

Upon admission to the VTH, during the first clinical examination, a blood sample was usually collected from the jugular or cephalic vein of each animal under sedation or general anaesthesia to reduce the stress and handling time [75]. The blood sample was immediately used for routine laboratory analyses in order to perform a rapid diagnosis, while an aliquot was collected separately in a serum tube and centrifuged within 30 min after collection at 1000 rpm for 10 min. The serum sample obtained was stored at −20 °C to be used for future analyses.

A faecal sample was also usually collected from each animal after natural voiding; an aliquot of the faecal sample was immediately stored at +5 °C and examined within 12 h for routine parasitological examinations (passive flotation, Baermann technique, coproantigen detection) while an aliquot was stored at −20 °C pending further analyses.

Therefore, all the serological and molecular analyses on serum and faecal samples were performed after the period of hospitalisation and on stored frozen samples.

### 2.4. Viral Pathogens

A species-specific serological and molecular analysis panel was designed in agreement with previously reported data in the available literature on the most commonly detected viral agents in the enrolled species [28,33,76,77,78,79,80,81,82,83,84,85]. Molecular analyses were performed on faecal samples.

#### 2.4.1. Serological Analysis

Roe deer (*n* = 23) and fallow deer (*n* = 4) sera were tested for the detection of antibodies against Bovine Viral Diarrhea Virus (BVDV) and Small Ruminant Lentivirus (SRLV), and all animals’ (*n* = 50) sera were tested for antibodies detection against Hepatitis E virus (HEV).

BVDV and SRLV antibodies were detected using the BVDV p80 Ab (IDEXX, Westbrook, ME, USA) and by Eradikit^®^ SRLV Genotyping Kit (IN3 diagnostic, Grugliasco, Italy), respectively, after adaptation. The SRLV kit is suitable for the detection and genotyping of SRLV antibodies as it can discriminate whether a positive sample belongs to genotype A (Maedi Visna-like), B (CAEV-like), or E (Roccaverano strain).

HEV antibodies were tested in all enrolled animals using the HEV Ab version ULTRA kit (DIA.PRO, Sesto San Giovanni, Italy). Optical density was measured using a plate reader (Multiscan FC; Thermo Scientific, Waltham, MA, USA) at a wavelength of 450 nm. The procedures and interpretation analysis of the analyses were carried out according to the manufacturer’s instructions.

#### 2.4.2. Molecular Analysis

Species-specific panels for molecular analysis are summarised in the Appendix A. For species with little data in the literature about previous virological studies, such as porcupine and pine marten, large panels were developed that included viruses identified in related species.

Due to the multiple species tested in this study and the paucity of data on virus circulation in wildlife, broad-spectrum PCR protocols with consensus primers covering as many viral species as possible were used.

Molecular analyses were performed on DNA and RNA extracted from 200 mg of faecal samples using the AllPrep PowerFecal DNA/RNA Kit (Qiagen, Hilden, Germany) according to the manufacturer’s instructions. DNA and RNA were eluted in 30 µL of Buffer AE and RNase-free water, respectively, and stored at −80 °C until analysis. PCR for the detection of Canine bocavirus-2 (CboV-2), Feline/canine parvovirus (FPV/CPV), Torque teno virus 1 and 2 (TTV-1, TTV-2), and Canine adenovirus (CadV), and both rounds of nested PCR for CboV-1 and 3 were performed on DNA using the HotStarTaq^®^ Plus Master Mix Kit (Qiagen, Germany). RT-PCR for the detection of *Canine distemper virus* (CDV), *Bovine viral diarrhea virus* (BVDV), *Bopivirus* (BoPV), and *Kobuvirus* (KoV), and first rounds of the nested RT-PCR for *Coronavirus* (CoV) and *Astrovirus* (AstV) were performed on RNA using the OneStep RT-PCR Kit (Qiagen, Germany). Secondary rounds were performed using the HotStarTaq^®^ Plus Master Mix Kit (Qiagen, Germany). The protocol used for CadV detection is reported to discriminate between CadV 1 and CadV 2, resulting in distinct amplicons of 508 bp and 1030 bp, respectively [86]. Samples positive at molecular analysis were subjected to sequence analysis (BMR genomics, Padova, Italy) to confirm PCR results and obtain phylogenetic information. The primer sets used in the PCR and rt-PCR were included as Appendix A [20,77,85,86,87,88,89,90,91,92,93,94,95]. PCR was performed according to the manufacturer’s instruction.

### 2.5. Parasitological Investigations

#### 2.5.1. Faecal Examinations

A passive flotation technique was carried out on the refrigerated aliquots to detect nematode eggs and coccidian oocysts, using a commercial saturated solution of sodium chloride (specific gravity = 1.20) as flotation fluid (Coprosol^®^, Candioli Farmaceutici, Beinasco, TO, Italy). The parasite burden was estimated by determining the number of eggs per gram (EPG) and oocysts per gram (OPG) of faeces by the modified McMaster method, with the lowest detection limit of 50 EPG/OPG per gram. In addition, faecal samples were examined via the Baermann technique to detect bronchopulmonary and cardiopulmonary nematode larvae. The number of larvae per gram (LPG) of faeces was estimated as above. The mean and range of OPG, EPG and LPG were calculated.

#### 2.5.2. Commercial Assays

A panel of endoparasite infections was investigated by rapid in-clinic tests in a different range of species. In particular, a rapid diagnostic test to detect both *Cryptosporidium* and *Giardia* coproantigens (Fastest^®^ Crypto-Giardia Strip, Megacor Diagnostik, Austria) was performed on all 47 faecal samples examined for parasitological infections. Moreover, sera from foxes (*n* = 10) and badgers (*n* = 5) were screened for the presence of antibodies against *Leishmania infantum* (Snap *Leishmania*, Idexx, USA), *Angiostrongylus vasorum* (Angio Detect Test, Idexx, Westbrook, ME, USA), and *Dirofilaria immitis* (Snap Heartworm Antigen Test, Idexx, Westbrook, ME, USA). Screening of seropositivity for *Neospora caninum* (Fastest^®^
*Neospora caninum* test kit, Megacor Diagnostick, Gemeinde Hörbranz, Austria) was carried out on foxes, badgers, roe deer (*n* = 23), and fallow deer (*n* = 4). All the rapid in-clinic tests were performed according to their respective manufacturer’s instructions. Finally, serum samples from all animals were screened for *Toxoplasma gondii* infection by a commercial ELISA kit (ID Screen^®^ Toxoplasmosis Indirect Multi-species, IDVET, Montpellier, France) according to the manufacturer’s protocol.

#### 2.5.3. Transtracheal Wash Technique

In 4/13 (30.8%) roe deer that tested positive for lungworm larvae via the Baermann technique (see above), a transtracheal wash (TTW) was performed immediately after humane euthanasia due to severe injuries. Briefly, a polyurethane catheter (14G) (Terumo, Roma, Italy) was introduced 5 cm distal to the larynx at 45° to the skin between two tracheal rings identified by palpation. Once the catheter entered the tracheal lumen, the needle was retracted, and the stylet advanced. Then a catheter (1.3 mm × 50 cm) (Buster dog catheter, Kruuse, Denmark) was passed through the stylet, down the trachea, and towards the lungs, until resistance was found. A bolus of 30 Ml of sterile saline was injected using a 60 Ml syringe and the fluid was immediately aspirated to recover at least 10 Ml of fluid [96]. When the aspiration of fluid was complete, the catheter and stylet were withdrawn simultaneously. The TTW fluid was split into two 4 Ml EDTA tubes for parasitological examination. When lungworm specimens were detected in the amount of fluid recovered, they were stored for 24–48 h in physiological saline at 4 °C before the morphological examination and then stored in 70% alcohol pending molecular procedures for identification to species level.

#### 2.5.4. Molecular Analysis for *Giardia duodenalis* Genotyping and *Cryptosporidium* Species Identification

Molecular investigation of *G. duodenalis* and *Cryptosporidium* was performed on faecal samples testing positive for coproantigens test (Fastest^®^ Crypto-Giardia Strip, Megacor Diagnostik, Gemeinde Hörbranz, Austria). Genomic DNA extraction was performed by QIAamp DNA Stool Mini Kit (QIAGEN, Hilden, Germany) according to the manufacturer’s instructions. Giardia genotyping was determined through the amplification of *tpi* and *β-giardin* genes. In detail, a 530 bp fragment of the *tpi* gene was obtained via a two-step nested PCR protocol using AL3543/AL3546 primers for primary PCR and AL3544/AL3545 for secondary PCR [97]. A 384 bp fragment of the *β-giardin* gene was amplified with a forward primer G7, a forward inner primer G376, and a reverse primer G759 [98].

*Cryptosporidium* genotyping was determined through the amplification of 800–850 bp of the GP60 gene. In a two-step nested PCR protocol, AL3531/AL3535 primers were utilized for primary PCR and AL3532/AL3534 for the secondary PCR [99].

All amplicons obtained were purified using the mi-PCR Purification Kit (Metabion International AG, Planegg, Germania), and sequencing was performed in the laboratory of Bio-Fab Research (Rome, Italy). In order to check the quality of the sequences, the resulting chromatograms were manually analyzed using Finch TYV1.4 software (Geospiza, Inc, Seattle, WA, USA).

Consensus sequences were then compared with those available in the GenBank database by using the Standard Nucleotide BLAST search and aligned by Clustal Omega implemented in MEGA X with representative sequences of *tpi*, *bg*, and *GP60* loci and used as references.

#### 2.5.5. Molecular Identification of Adult Lungworms

Adult lungworms collected by TTW were identified to species level by PCR technique. Molecular analyses were performed by BMR Genomics (Padua, Italy, https://www.bmr-genomics.it/ accessed on 7 January 2022). Genomic DNA was extracted from the stored adult lungworms using the commercial PCRBIO Rapid Extract Lysis Kit (PCR Biosystems) following the manufacturer’s instructions. A fragment (about 700 base pairs in length) of the mitochondrial gene for cytochrome C oxidase subunit 1 (COX1) was used as a DNA barcoding system and amplified. The PCR amplification was carried out in a final mixture containing 12.5 μL of AmpliTaq Gold™ 360 Master Mix (Thermo Fisher), 2 μL of gDNA extracted, 1 μL (10 μM) of LCO1490 primer (5′-GGTCAACAAATCATAAAGATATTGG-3′), 1 μL (10 μM) of HCO2198 primer (5′-TAAACTTCAGGGTGACCAAAAAATCA-3′), and deionized water, reaching a total reaction volume of 25 μL. The PCR reactions were subjected to the following conditions in a thermal cycler (Mastercycler^®^, Eppendorf): 95 °C × 2 min, then 5 cycles (95 °C × 40 s, 45 °C for 90 s, and 72 °C for 90 s), followed by 35 cycles (95 °C for 40 s, 50 °C for 90 s, and 72 °C for 60 s), and finally, 72 °C for 10 min. The amplification products were visualized after electrophoresis on 1.5% agarose gel. PCR products were purified with ExoSAP-IT™ (Thermo Fisher Scientific, Massachusetts, MA, USA), sequenced, and aligned via BLAST analysis to detect their identity by retrieving similar sequences deposited in NCBI’s GenBank database [100,101].

### 2.6. Statistical Analysis

The prevalence of virological and parasitic agents was determined as the number of positive animals/numbers of examined animals X 100.

The Fisher test was used to determine statistical differences between rescue areas on positivity/negativity for *Astrovirus*, *Kobuvirus*, *Bopivirus*, and *Adenovirus*.

## 3. Results

### 3.1. Viral Pathogens

#### 3.1.1. Serological Analysis

One out of 23 roe deer (4.3%) scored positive for antibodies against BVDV and one for SRLV, genotype A Maedi Visna-like, while all the fallow deer were serologically negative for both viruses. The positive roe deer was a male rescued in zone B. All the animals (50/50, 100%) scored negative for HEV antibodies.

#### 3.1.2. Molecular Examination

All animals tested negative for CBoV, FPV, CPV, TTV, BVD, CDV, and CoV. Animal distribution in the three-rescue area and results of virological molecular analysis presented by viruses, species and rescue area are reported in Table 3 and Table 4.

Six out of 50 animals (12%) tested positive for KoV: 1/23 roe deer; 2/12 foxes; 2/4 porcupines; and 1/6 badgers, respectively. All positive roe deer, badger, foxes, and one porcupine were male; the other porcupine was female. All positive animals belonged to zone A; thus, no statistical comparison among zones was performed.

Eight out of 50 (16%) animals tested positive for AstV: 3/23 roe deer (1 male and 2 female) and 5/12 foxes (4 male and 1 female). Four out of 8 (50%) animals belonged to zone A (3 foxes and 1 roe deer), 3 (37.5%) animals belonged to zone B (2 foxes and 1 roe deer) and 1 (12.5%) roe deer belonged to zone C. No differences were found between zone A vs. B (*p* = 0.626), zone A vs. C (*p* = 0.536), or zone B vs. C (*p* = 0.779).

Foxes (*n* = 12), badgers (*n* = 6), and pine martens (*n* = 1) were tested for CAdV, and 5 out of 19 (26%) animals tested positive for CAdV type 1. Three out of 12 foxes and 2/6 badgers were positive; all the positive foxes were male, 2 from zone B and 1 from zona A. Both positive badgers were male, 1 from zone A and 1 from B. No animals tested for CAdV have been rescued in zone C. No statistical differences were found in CAdV type 1 prevalence between zones A vs. B (*p* = 0.161).

Roe deer (*n* = 23) and fallow deer (*n* = 4) were tested for BoPV, and 1 out of 27 (3%) tested positive for BoPV. This was a female fallow deer from zone A; thus, no statistical comparison among zones was performed.

Appendix A shows the accession numbers of the sequences entered in the GenBank database, along with the results of their analysis with Blast software. For each sequence in this study, the analysis identified a reference sequence and determined the per cent of nucleotide identity and E value obtained from their comparison.

### 3.2. Parasitological Investigations

Overall, parasitological analyses were performed in a total of 47/50 (94%) animals, including 10 foxes, 5 badgers, 4 porcupines, 23 roe deer, 4 fallow deer, and 1 marten, as faecal and serum samples from 2 foxes and 1 badger were not available. Results of faecal examinations by flotation and Baermann techniques are shown in Table 5.

#### 3.2.1. Faecal Examinations

Roe deer. Overall, 19/23 (82.6%) roe deer were found to be coprologically positive by flotation and/or Baerman techniques. Ten out of 23, 8/23, and 1/23 roe deer showed two, one, or three parasitological infections, respectively. Eggs of gastrointestinal strongyles (mean = 7, range ≤ 50–100 EPG), *Eimeria* oocysts (mean = 250, range ≤ 50–>1000 OPG), and *Capillaria* spp. eggs (mean ≤50 EPG), were found in 14/23, 5/23, and 2/23 roe deer, respectively. 

Lungworm larvae (mean = 23, range ≤ 50–150 LPG) consistent with first-stage larvae of *Dictyocaulus* sp. were detected in 13/23 (56.2%) roe deer.

Red foxes. All foxes (10/10, 100%) were found to be parasitised. Two, three, four, and even six parasitic co-infections were detected in 5/10, 2/10, 1/10, and 1/10 foxes, respectively. Therefore, only one fox out of 10 showed a single parasitic infection with coccidia oocysts. Parasite prevalence was as follows: *Ancylostomatidae* eggs (mean = 344, range ≤ 50–>1000 EPG) were found in 8/10 foxes; *Toxocara canis* eggs (mean ≤50 EPG) in 5/10; *Capillaria* spp. eggs (mean = 75, range ≤ 50–300 EPG) in 4/10; *Trichuris vulpis* eggs (mean ≤50 EPG) in 3/10; and coccidia oocysts (mean = 600, range = 200–>1000 OPG) in 2/10. In addition, the spurious passage of *Hymemolepis* eggs (<50 EPG) was detected in 1/10 foxes. Larvae of *A. vasorum* (mean = 275, range < 50–>1000 LPG) were detected in stool samples from 4/10 foxes.

Badgers. All badgers (5/5, 100%) harboured parasites. Infections with two and four parasitic agents or only a single infection were detected in 3/5, 1/5, and 1/5 of them, respectively. *Ancylostomatidae* eggs (mean = 80, range ≤ 50–300 EPG) and larvae of *Crenosoma* sp. (mean = 137, range ≤ 50–400 LPG) were found in stool samples from 5/5 and 4/5 badgers, respectively, while *Strongyloides* eggs (<50 EPG), *Capillaria* spp. eggs (<50 EPG), and coccidia oocysts (<50 OPG) were detected in 1/5 each.

All the fallow deer, all the porcupines, and the marten were found negative for both the flotation and Baermann techniques.

#### 3.2.2. Commercial Assays

Results of the rapid in-clinic tests and the ELISA kit are summarized in Table 5. Briefly, eleven, six, and two samples from different host species showed positive results for *N. caninum*, *G. duodenalis*, and *C. parvum*, respectively. Six and four samples from foxes tested positive for *A. vasorum* and *D. immitis*, respectively, and one sample from badgers tested positive for *D. immitis*. No samples tested positive, either for *L. infantum* or *T. gondii*.

#### 3.2.3. TTW Results

Four adult lungworms were detected in the TTW fluid from one of the four (25%) roe deer found positive for *Dictyocaulus* sp. larvae via the Baermann technique. The specimens measured on average about 5 cm in length and 0.4 mm in width at the mid-body and had an elongated oral opening, dorso-ventrally flattened. Based on the length of the body and the shape of the oral opening, the lungworms were presumptively identified as *Dictyocaulus* sp.

#### 3.2.4. Molecular Identification of *Giardia duodenalis* and *Cryptosporidium* spp.

A total of six samples for *G. duodenalis* and two for *Cryptosporidium* spp., carried out positive by the coproantigens test, were tested by the two-step nested PCR protocols. By sequencing at the tpi locus, 1/6 (16.7%) sample from a roe deer and 1/6 (16.7%) from a porcupine were assigned to Assemblage A-AI and Assemblage B-BIV, respectively. Sequences obtained in this study were deposited in the GenBank database and are available under the accession numbers OP946512 and OP946513, respectively.

#### 3.2.5. Molecular Identification of Adult Lungworms

The BLAST search results showed that COX1 of adult lungworms collected from a euthanized roe deer had 99–100% DNA homology with sequences of *Dictyocaulus capreolus* available in the GenBank database. The sequence obtained in this study was deposited in the GenBank database and is available under the accession number: OQ209836. To the best of our knowledge, this is the first molecular identification of *D. capreolus* in roe deer in Italy.

## 4. Discussion

Serological surveys in wildlife are a diagnostic tool to demonstrate the presence of antibodies against specific pathogens and to provide epidemiological data on pathogens; however, serological sampling in wild animals is not always feasible and can be very stressful for the animals themselves. In the present study, the hospitalisation of wild animals allowed blood sampling during clinical practice, and we decided to investigate by serological analysis a panel of pathogens for which direct virological diagnosis on faecal samples is not applicable.

The detection of BVDV antibodies indicates the presence and circulation of the virus in wild animals; moreover, the absence of vaccination plans for wildlife removes doubts about possible vaccine seropositivity. In our survey, one roe deer out of 21 was serologically positive. This result is in agreement with data from European studies, which reported the absence of seropositive animals (Spain and England) up to 12% of seroprevalence (Norway) [14,102,103,104,105]. In Italy, a serosurvey was carried out in four large alpine ungulates in the High Valley of Susa, and antibodies were detected in chamois, wild boar, and red deer, but all roe deer were negative [15]. A similar result was obtained a few years later when Fernández-Sirera and colleagues tested domestic ruminants, cattle, sheep, wild ruminants, chamois, and roe deer in the same research area for the presence of antibodies against BVD. High prevalence was found in both domestic ruminants, 25% in cattle, 17% in sheep, and 42 % in chamois, but none of the 213 roe deer tested positive [16]. The results of the Italian studies and the low seroprevalence found throughout Europe, even in areas with regular contact with domestic animals with high prevalence, suggest that roe deer play a marginal role in the epidemiology of the virus and that they are mostly an occasional guest and does not represent a major risk to domestic ruminants. The low prevalence of the virus among the European roe deer populations compared to other wild deer, especially the chamois, is explained by the different habits of the roe deer. The roe deer is mainly a solitary animal that prefers bushy feeding areas to open pastures where it would be easy to encounter domestic ruminants [15]. In addition, it is possible that this species has low susceptibility to common pestiviruses. Indeed, Fischer and colleagues have discovered a new pestivirus subtype in roe deer [106], and this may influence the serological results. However, as the presence of BVDV infection in roe deer is rarely detected, it would be advisable to conduct further surveys in the study area to increase the number of samples and obtain a more reliable estimate of prevalence.

In addition, 1 roe deer tested positive for SRLV, genotype A Maedi Visna-like. Although SRLV were originally thought to be species-specific pathogens, several lines of evidence showed that lentiviruses can efficiently cross the species barrier and adapt to new hosts [107,108]. Several studies reported the detection of SRLV antibodies in wild ruminants: Rocky Mountain goats; red deer; roe deer; and mouflon [109,110,111]. Therefore, it is possible that contact between wild and domestic small ruminants could result in cross-transmission of SRLV.

Regarding viral molecular analysis, positivities in foxes (CAdV-1, KoV, AstV), roe deer (KoV, AstV,) fallow deer (BoPV), badgers (CAdV-1, KoV), and porcupine (KoV) have been detected.

A recent investigation in Italy revealed the presence of *bopiviruses* in sheep, roe deer, chamois, and Alpine ibex, with sequences clustering with *bopivirus* strains previously detected in goats and sheep from Hungarian farms and in red deer with sequences closely related to *Bopivirus C* sequences identified in fallow and red deer in Australia [112]. In our study, one fallow deer sample resulted positive for *Bopivirus*, whereas all samples from roe deer were negative. According to BLAST analysis, the sequence shared 94–92% nucleotide (nt) identity with deer *Bopivirus* strain (*Bopivirus C*), 88–85% with bovine *Bopivirus* strain (*Bopivirus A*), and 72–70% with ovine–goat *Bopivirus* strain (*Bopivirus B*). The sample, therefore, showed the highest nucleotide identity to *Bopivirus C* previously identified in Australian fallow deer and Italian red deer. Although numerous viruses of the Picornavidae family are responsible for diseases with relevant clinical forms in humans and animals, the pathogenicity of *Bopivirus* in deer remains undetermined. The population In our study, as in previous studies, was apparently healthy or not affected by symptoms attributable to an infectious disease.

*Astroviruses* were detected in foxes and roe deer representing, to our knowledge, the first report of *astrovirus* in these two species in Italy. *Astroviruses* have already been identified in foxes in the Netherlands and Australia using metagenomic techniques. In the first study, the sequences were mainly identified as two novel *astroviruses* called Fox *Astrovirus* F5 and F4, in the second, sequences clustered with feline *astrovirus* with high nt identity. Little research has been reported on AstVs in deer, but *astroviruses* have been identified in European roe deer in Denmark in 2010 [113] from animals with gastrointestinal disease, and in Slovenia from healthy hunted animals [114]. The following sequence analysis identified a new species of *Mamastrovirus*, the *Capreolus capreolus Astrovirus* (CcAstV strain 1 and 2), related to other bovine, porcine, yak, porcupine, and dromedary AstVs strains, possibly forming a group of currently unassigned sequences that are distantly related to mAstV genogroups I and II. Our sequences, both from foxes and roe deer, were obtained using broad-range primers amplifying a portion of RdRp without phylogenetic information; however, BLAST analysis of the amplicons confirms that this virus clustered with other mammalian-associated viruses within the *mamastrovirus* genera. In general, AstV infection is associated with mild to severe gastrointestinal tract disease, typically with diarrhoea and vomiting [115], but has also been associated with other diseases, such as shaking syndrome in mink, neurological disease in cattle and sheep, and encephalitis in humans [116,117,118,119]; infection may be asymptomatic [115]. There is no information on the symptoms in foxes, although it is likely that CcAstV can cause gastroenteritis in roe deer. In our study, the sampled animals did not have gastrointestinal disease or other disorders attributed to *astroviruses*.

In Italy, *kobuviruses* have been detected in dogs, foxes, wolves, roe deer, goats, swine, cats, and cattle [33,38,39,40,76,120]. In this study, *kobuviruses* were detected in two foxes, one roe deer, one badger, and two porcupines, and none of the positive animals showed symptoms suggestive of enteritis. In foxes, Canine *kobuvirus* (CaKoV) has been detected in free-ranging foxes in Italy with a prevalence of 14.7%. To date, the known host range of CaKoV includes several species of Canidae (domestic dog, red fox, wolf, golden jackal, and side-striped jackal), and the virus has been detected in both diarrhoeic and asymptomatic animals, so the pathogenic potential of *kobuviruses* in carnivores remains to be elucidated. Our sequence of RdRp partial region from fox shared the highest identity with the CaKoV strains from Italian foxes with an nt sequence identity of 89–94% showing E-value ranging from 2e^−30^ to 9e^−39^. CaKoV is often found in co-infections as a potential secondary infection after primary infections by mainly viral agents responsible for immunosuppression, such as canine distemper virus or parvovirus [121,122,123]. In the studies of Di Martino and colleagues, both a fox and some dogs were co-infected by *Kobuvirus* and cCoV [40].

In our study, all foxes were negative to molecular analysis for CPV, CDV, and cCoV, but two foxes positive for *kobuvirus* were also PCR-positive for *astrovirus*, and one also for adenovirus; both were responsible for clinical gastroenteritis. Coinfection with CaKoV and cAstV and with CaKoV and cAdV has been described in diarrhoeic dogs in China and Korea, respectively [124,125]. In roe deer, *kobuvirus* RNA was found in 6.6% of rectal swabs collected from healthy roe deer in northern Italy between 2012 and 2014; most of these strains showed high homology with bovine *kobuvirus*, and others with caprine *kobuvirus* [76]. In our study, *kobuvirus* has been detected in 1/23 roe deer with a prevalence in line with previous studies. BLAST analysis of the partial 3D gene sequence showed 75% and 71% of nucleotide identity, respectively, with bovine and caprine *kobuvirus* detected in Italian roe deer, while the highest identities were shared with Italian fox *kobuvirus*. This result contrasts with what resulted in the previous study and it cannot be excluded that the roe deer may have ingested food contaminated with faecal contents of other species. However, it is important to consider that this is the second study to report *Kobuvirus* in wild ruminants and that very little is still known about which strains circulate among the populations of these animals and about the possibility of transmission of strains between different species. *Kobuvirus* has only been identified in one member of Mustelidae, a domestic ferret in the Netherlands, using a metagenomic approach, and the almost complete *kobuvirus* genome clustered with bovine and ovine *kobuvirus* (*Aichivirus B*) possibly indicates a past cross species transmission events [126]. No data is currently present about the circulation of *kobuvirus* in free-ranging mustelids that share the ecosystem with other wild animals susceptible to infection. In the present study, *kobuvirus* has been identified in 1/6 badger representing its first detection in this species. The badger resulted also positive for adenovirus. Our sequence clustered with sequences of CaKoV detected in wild and domestic carnivores with an nt sequence identity of 85–87% and an e value of 3 e^−80^–1 e^−65^. Finally, in our study, *kobuvirus* was also identified in two porcupines representing its first detection in this species. To date, *kobuvirus* has been detected in other rodents, such as rabbits, grey squirrels, and some species of mice (Peromyscus crinitus, Apodemus agrarius, Rattus norvegicus, Rattus losea, and Rattus argentiventer, Mus musculus, Rattus losea, Rattus tanezumi, and Rattus norvegicus) [30,36,127,128,129]. The BLAST analysis of our sequence showed a high homology both with both CaKoV (90–95% of nt identity, e value of 9 e^−98^–2 e^−42^), and murine *kobuvirus* (85–90% of nt identity, e value of 6 e^−81^–1 e^−64^), both belonging to the same genetic cluster within Aichivirus A, and more in-depth phylogenetic studies would be needed to better classify the virus. Interestingly, a high percentage of nt identity resulted between all our sequences (87–95%). This evidence, together with the common origin of all positive animals from zone A, suggests the possibility that a similar *kobuvirus* could infect different animal species sharing the same habitat.

Moreover, three foxes and two badgers tested positive for cAdV-1. In foxes, cAdV was first identified in 1925 as the causative agent of severe neurologic disease, the epizootic fox encephalitis [130]. Subsequently, evidence of cAdV circulation in foxes has been reported in several geographical areas, with serological prevalence in the European population ranging from 3% in Germany to 64.4% in the UK [131,132]. In Italy, and more specifically in the Pisan area, the virological prevalence reported in previous studies is around 28% [132], which is in line with the value of 25% obtained in our study. All the positive animals were traumatised, and none showed clinical signs attributable to infectious diseases. Our data, together with the high prevalence values reported worldwide [44,45,132,133,134] and the identification of the virus in healthy animals reported in previous studies [132,134], suggest that red foxes may play a role in maintaining cAdV-1 in the territory and may be a source of viral spread in wild ecosystems. In addition, although cAdV-1 can be controlled in domestic dogs by vaccination, cases of infectious canine hepatitis in domestic dogs have been diagnosed in Italy in recent years [135], and the presence of infected foxes may also be a source of infection for domestic dogs, especially foxes living in peri-urban areas [135]. Adenovirus has previously been detected in Mustelidae, in a young striped skunk with hepatitis, in a seriously diseased Eurasian otter in the Seoul Grand Park Zoo, and in several pine martens and Eurasian otters in the UK, with prevalences of 33% and 88%, respectively [80,136,137]. However, our evidence is the first detection of adenovirus in badgers.

Both fox and badger viral sequences clustered with *Canine adenovirus 1* with a high percentage of nt identity (94–98%). The presence of cAdV-1 in badgers was expected as specific adenoviruses have also been identified to date in otters and pine martens (marten adenovirus and lutrine adenovirus) and it is already known that mustelids are susceptible to CAdV-1 infection [80]. Moreover, the home ranges of mustelids and other predators such as red foxes (*Vulpes vulpes*) may overlap and there is potential for cross-infection with pathogens through indirect contact with urine, faeces, and other infected fomites.

Moving on to discuss the parasitological findings, although coccidiosis was presumably harmless for the animals examined in this study, coccidia may pose a major threat to some species of wildlife; for instance, juvenile coccidiosis may be associated with impaired growth and increased mortality in badger cubs [50]. In this study, oocysts of *Eimeria* were detected in fecal samples from roe deer, red foxes, and badgers, but identification to species level could not be reached.

Normally GSI are of little clinical significance in wild ruminants, but if combined, they can cause profuse and watery diarrhea, anaemia of varying degrees, edema under the lower jaw extending along the abdomen, progressive weight loss, rough hair coat, anorexia, hypoproteinemia, reduced growth, and even death in severe cases [138].

Since a much higher prevalence of *C. bovis* (26.6%) than *C. bilobata* (6%) has been detected in *C. capreolus* [52,53], it is likely that *C. bovis* was the *Capillaria* species involved in roe deer of this study.

Adults of *D. capreoli* were detected in the transtracheal wash fluid from a roe deer, as shown by the results of the molecular analysis. *D. capreoli* has been reported to cause anorexia, poor haircoat quality, delayed haircoat change, diarrhea, small apical atelectatic areas, peri-bronchial inflammation, and total bronchopneumopatia in roe deer [55].

In agreement with previous studies carried out in Italy [56,57], combining results of coprological analyses and commercial assays, intestinal worms, lungworms, and heartworms were found in red foxes. It has been reported that the infection with intestinal helminths, especially *T. canis*, and the number of helminth species per individual may negatively affect the host body condition in red foxes [139,140]. All parasites of foxes are shared with dogs [141], and some species can be zoonotic agents [142]. In particular, foxes may be considered an important source of infections with cardiopulmonary nematodes, which may be responsible for overt clinical disease in dogs [143,144].

The eggs of *A. caninum* and *U stenocephala* cannot be reliably distinguished microscopically. Likewise, eggs of *C. aerophila* and *C bohemi* are microscopically indistinguishable from one species to another because all *Capillaridae* eggs are morphologically characterized by a similar barrel shape with polar plugs [134]. Therefore, it was not possible to determine whether the foxes examined in this study were infected with A. *caninum, U*. *stenocephala*, or both and with *C. aerophila*, *C. boehmi*, or both. In addition, the spurious expulsion of *Capillaria hepatica* (syn. *Calodium hepaticum*) eggs, after preying or scavenging on infected hosts, cannot be excluded. Indeed, foxes are known to scavenge on carcasses and eat a wide range of rodents, including mice and rats, which commonly show a very high prevalence of infection with *C. hepatica* [145]. In this study, the spurious expulsion of typical eggs of *Hymenolepis* spp. was detected. *Hymenolepis* is a genus of zoonotic cestodes that commonly infect rodents, mostly mice and rats, with usually a high prevalence [146]. This suggests that at least some of the foxes examined were feeding on rodents at the time of sampling and that foxes may act as carriers for the dispersal of the eggs of *Hymenolepis* species into the environment. Conversely, though *Capillaria* plica may be highly prevalent in foxes in central Italy [147], shedding of C. plica eggs in faeces was excluded because adult parasites of *C. plica* live in the bladder of definitive hosts, and eggs are expulsed with urine.

Within the family, *Ancylostomatidae U. criniformis* is the species widely reported in European badger populations [59,61,62,63]. Therefore, it may be suggested that *U. criniformis* was the species infecting badgers examined in this study.

Lungworm infections in M. meles can be identified based on the typical morphology of L1 larvae (*Crenosoma* spp.) and eggs (*C. aerophila*). However, the limited information available on the morphological differences between L1 larvae of *C. melesi* and *C. vulpis* did not allow differentiation from one species to another. Therefore, it is likely that *C. aerophila* was the *Capillaria* species involved in the badgers of this study, but it was not possible to determine whether *C. melesi*, *C. vulpis*, or both were infecting the badgers examined. Similarly, to foxes, the shedding of spurious *C. hepatica* eggs, merely passing through the intestinal tract, cannot be completely excluded. Indeed, *C. aerophila* and *C. hepatica* eggs are hardly distinguishable by microscopy, while badgers are known to be scavenger animals and to eat small mammals, including mice and rats (often heavily infected by *C. hepatica*, as aforementioned).

Oval, thin-shelled eggs with a fully developed coiled larva inside (typical of rhabditiform nematode parasites) were found in a badger faecal sample and were identified as eggs of *Strongyloides* sp. To our knowledge, the identification of *Strongyloides* species infecting the European badger is still unknown since *Strongyloides* specimens recovered in M. meles have not been identified to the species level [61,62]. *Strongyloides* specimens isolated from the Japanese badger (*Meles anakuma*) are closely related to *Strongyloides procyonis* on a phylogenetic tree and are morphologically similar to *S. procyonis* and *Strongyloides martis* [65]. However, the results of molecular phylogenetic analyses using mitochondrial genome sequencing as a marker suggested that they belonged to a species genetically separate from *S. procyonis* [65]

Concerning *Leishmania* and *Toxoplasma*, it cannot be excluded that the negative results found in this study may be due to limitations (uncertain sensitivity) attributable to the screening tests. A rapid immunochromatographic test (marketed for dogs) has been previously used to determine the seroprevalence of anti-*Leishmania* antibodies in wild canids, such as red foxes and coyotes [148]. Likewise, a commercial indirect multi-species ELISA (marketed for ruminants, swine, dogs, and cats) has been previously used to determine the seroprevalence of anti-*Toxoplasma* antibodies in free-ranging and captive large African carnivores [149]. Therefore, at present, it can be assumed that these two commercial assays may be suitable for wildlife species, but they have not yet been validated for the diagnosis of *Leishmania* and *Toxoplasma* infections in the species examined in this study.

*Neospora caninum*-antibodies were detected in 3/4 fallow deer, 7/23 roe deer, and 1/5 badgers. These findings agree with those of other studies where the reported seroprevalence values for *Neospora* were 11% in fallow deer [150] and 28% in roe deer [151], while 10.9% of badgers were found positive for *Neospora* DNA in brain, liver, or neck muscles [152]. Red foxes are considered natural intermediate hosts for N. caninum, with reported prevalences of 10.7% by the PCR [153] to 69.8% by the agglutination test [154]. However, no foxes tested positive for *Neospora* in this survey. The possible existence of a sylvatic life cycle of N. *caninum* has been postulated [69]. Therefore, the present results suggest that fallow deer, roe deer, and badgers may contribute to the sylvatic life cycle of N. *caninum*, if any, in some areas of central Italy. Further investigations are needed to elucidate the role they may play in maintaining the circulation of the parasite in nature.

For the detection of *A. vasorum* infection in red foxes and badgers, the Angio Detect test was used in combination with the typical recovery of cardiopulmonary larvae by the Baermann technique. Angio Detect is commercialized as an in-clinic assay to detect *A. vasorum* circulating antigen in dogs but it has also been successfully used in other host species, such as red foxes and badgers [155,156]. Likewise, commercially available antigen detection tests, commercialized for use in dogs, have been successfully used to identify *D. immitis* infection in other host species, such as wolves and red foxes, coyotes, and island foxes [157,158,159]. In this study, the use of the Angio Detect rapid test enhanced the global sensitivity of the diagnostic procedure in foxes as it allowed the detection of two additional positive subjects concerning the traditional Baermann apparatus. It is likely that these two *A. vasorum*-antigen positive/Baermann negative foxes were in the prepatent period that characterises the early phase of any parasitic infection.

The European badger may be a definitive host for the cardiopulmonary nematodes *D. immitis*, *A. vasorum*, and *A. daskalovi* [61,62,66,67,68]. In this study, *D. immitis* circulating antigen was detected in 1/5 badgers by the in-clinic test, but no one was found positive either for circulating antigen or larvae of *Angiostrongylus* spp.

In this study, the molecular analysis of *G. duodenalis* was successfully amplified at the tpi locus in one roe deer and in one porcupine. Among isolates of *G. duodenalis* from *C. capreolus*, assemblage A was previously identified in the Netherlands and Poland, while assemblage B was detected in Poland [160,161,162]. Identification to sub-assemblage level was not carried out in those investigations. *G. duodenalis* sub-assemblage AII was reported in roe deer in Spain [163]. In the present study, sub-assemblage AI was identified. To the best of our knowledge, this is the first report of *G. Duodenalis* sub-assemblage AI in *C. capreolus* in Italy. This sub-assemblage has previously been isolated in other wild species in the same geographical area [164]. Likewise, the finding of *G. duodenalis* sub-assemblage BIV in a porcupine is consistent with data previously reported for the same host species in the same geographical area [165]. *G. duodenalis* sub-assemblages AI and BIV are mainly found in animals and, to a lesser extent, in humans. Therefore, they are considered potentially zoonotic [70].

## 5. Conclusions

The results highlight the widespread presence of pathogens in the population studied, confirming the role of wildlife in the epidemiology of several viruses and parasites. Although the impact on wild species’ health is not yet known for all the pathogens identified, they can pose a threat to wildlife and can potentially have important effects on the population dynamics of their hosts, especially when they are experiencing the cumulative effects of other pathogens and stressors.

Moreover, some of the parasitic agents found can be transmitted to domestic animals, and some of them are zoonotic. Therefore, they can affect the health status of livestock or pets and can have public health implications, especially in a highly urbanised area such as the study area.

The study highlights the key role that structures admitting wild animals can play as focal points for collecting information on the health of wildlife populations and monitoring the health of the environments in which they live.

## Figures and Tables

**Figure 1 animals-13-00931-f001:**
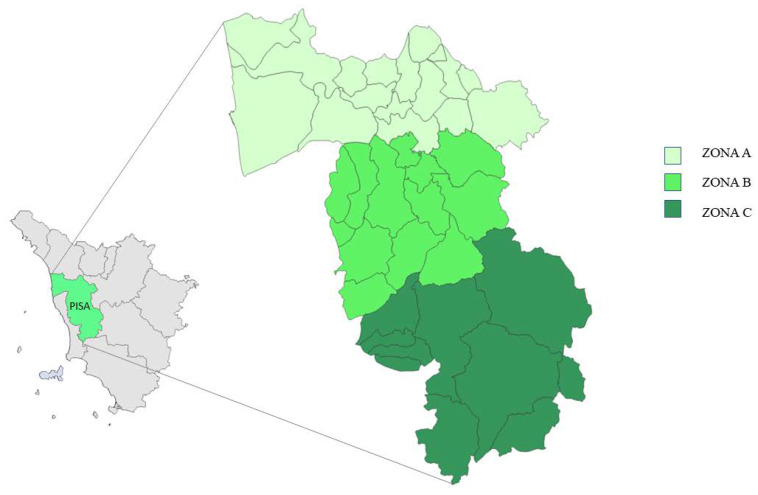
Map of the province of Pisa divided into the 3 zones.

**Figure 2 animals-13-00931-f002:**
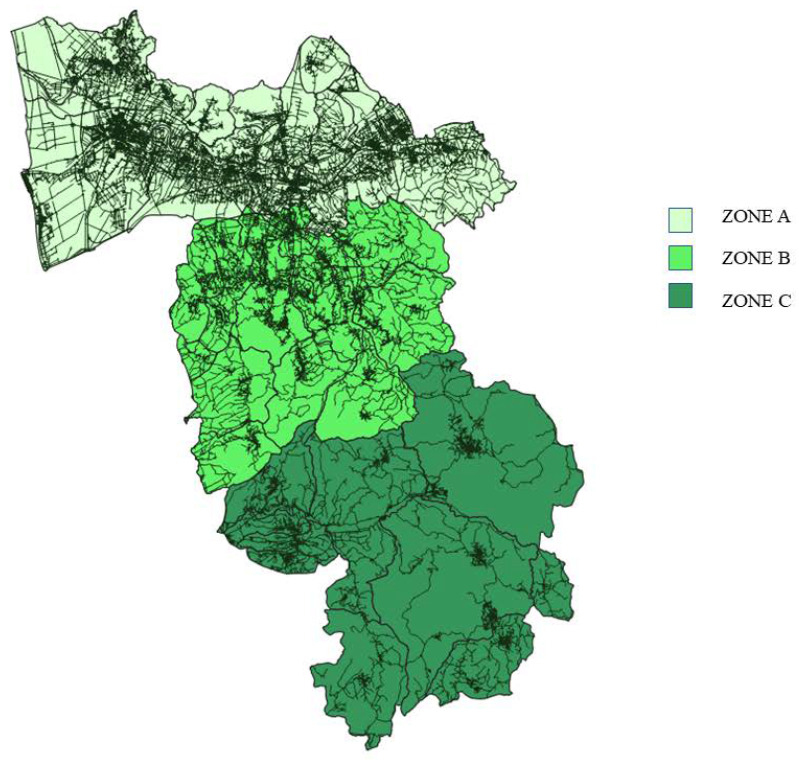
Development of the road network in the province of Pisa.

**Figure 3 animals-13-00931-f003:**
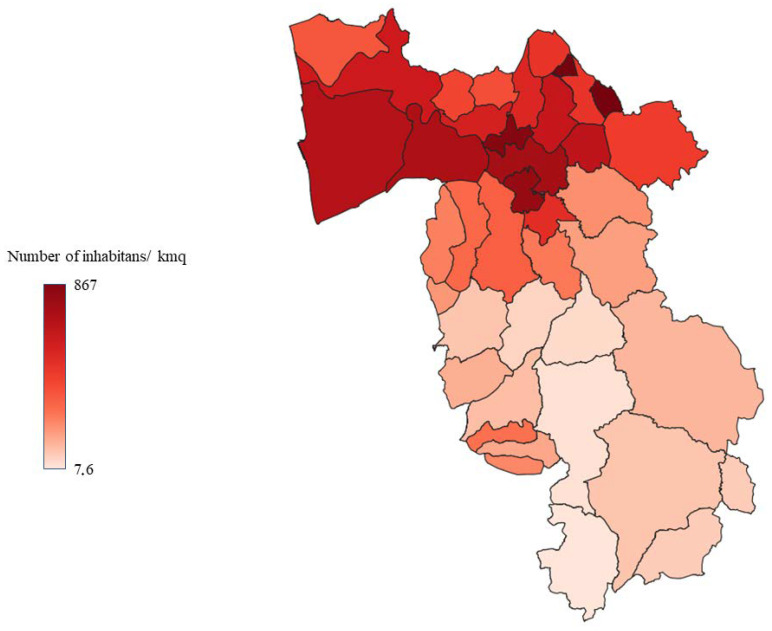
Municipalities in the province of Pisa classified according to population density.

**Figure 4 animals-13-00931-f004:**
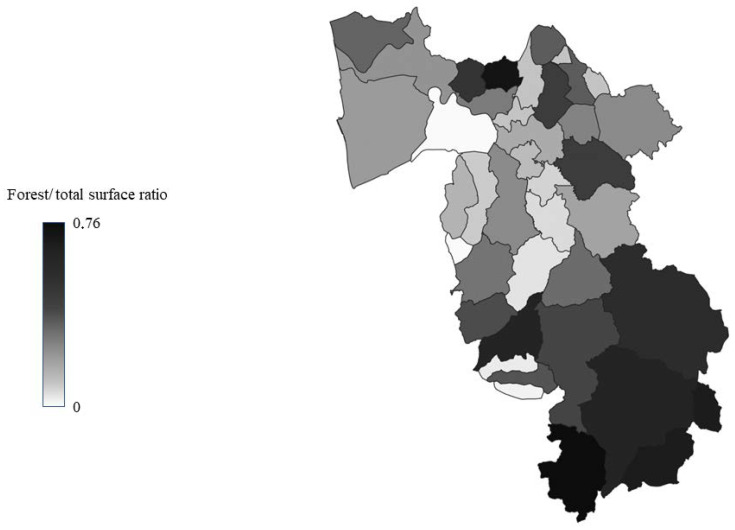
Municipalities in the province of Pisa classified according to the forest/total surface ratio.

**Table 1 animals-13-00931-t001:** Data relating to area extension, number of inhabitants, population density, and forest extension (expressed as its relationship with the total area surface) for each zone.

	Surface (kmq)	Population	Population Density	Forest/Total Surface Ratio
**ZONE A**	824.4	328,873	398	0.3
**ZONE B**	588.3	63,313	107	0.2
**ZONE C**	939.6	25,763	27	0.5

**Table 2 animals-13-00931-t002:** Number of enrolled animals presented by species and sex.

Species	Sex		Total
	Males	Females	
Roe deer	17/23 (74%)	6/23 (26%)	23
Fallow deer	3/4 (75%)	1/4 (25%)	4
Foxes	9/12 (75%)	3/12 (25%)	12
Badgers	4/6 (67%)	2/6 (33%)	6
Pine martens	-	1/1 (100%)	1
Porcupines	2/4 (50%)	2/4 (50%)	4
**Total**	**35/50 (70%)**	**15/50 (30%)**	**50**

**Table 3 animals-13-00931-t003:** Animal distribution in the three-rescue zones.

Species	Zone			Total
	Zone A	Zone B	Zone C	
Roe deer	9/23 (39%)	11/23 (48%)	3/23 (13%)	23
Fallow deer	4/4 (100%)	-	-	4
Foxes	9/12 (75%)	3/12 (25%)	-	12
Badgers	5/6 (83%)	1/6 (17%)	-	6
Pine martens	-	1/1 (100%)	-	1
Porcupines	3/4 (75%)	-	1/4 (25%)	4
**Total**	**30/50 (60%)**	**16/50 (32%)**	**4/50 (8%)**	**50**

**Table 4 animals-13-00931-t004:** Results of virological molecular analyses presented by viruses, species and rescue zones. A, B, and C refer to rescue zones.

	CAdV 1					KoV			AstV			BoPV		
**Roe deer**	--					**1**			**3**			**0**		
						*A*	*B*	*C*	*A*	*B*	*C*			
						1/9	0/11	0/3	1/9	1/11	1/3			
**Fallow deer**	--					**0**			**0**			**1**		
												*A*	*B*	*C*
												1/4	0/0	0/0
**Foxes**	**3**					**2**			**5**			--		
	*A*		*B*		*C*	*A*	*B*	*C*	*A*	*B*	*C*			
	1/9		2/3		0/0	2/9	0/3	0/0	3/9	2/3	0/0			
**Badger**	**2**					**1**			**0**			--		
	*A*		*B*		*C*	*A*	*B*	*C*						
	1/5		1/1		0/0	1/5	0/1	0/0						
**Pine** **Marten**	**0**					**0**			**0**			--		
**Porcupine**	--					**2**			**0**			--		
						*A*	*B*	*C*						
						1/3	0/0	0/1						
**Total**	**5**					**6**			**8**			**1**		
	*A*	*B*		*C*		*A*	*B*	*C*	*A*	*B*	*C*	*A*	*B*	*C*
	2/14	3/5		0/0		6/30	0/16	0/4	4/30	3/16	1/4	1/13	0/11	0/3

**Table 5 animals-13-00931-t005:** Parasite infections found by faecal flotation, the Baermann technique, and different commercial assays in the cohort of 47 wildlife animals examined. * GIS = Gastrointestinal strongyles; ^ ND = Not Determined.

Host Species	Faecal Flotation	Baermann Technique	Commercial Assays						
			*Giardia*	*Cryptosporidium*	*Toxoplasma*	*Neospora*	*Leishmania*	*Dirofilaria*	*Angiostrongylus*
Roe deer (*n* = 23)	* GIS (10) *Eimeria* (5) *Capillaria* (2)	*Dictyocaulus* (13)	1	0	0	7	ND	ND	ND
Fallow deer (*n* = 4)	Negative	Negative	0	0	0	3	ND	ND	ND
Red foxes (*n* = 10)	*Ancylostomatidae* (8) T canis (5) *Capillaria* (4) *T vulpis* (3) Coccidia (2) *Hymenolepis* (1)	*Angiostrongylus* (4)	1	1	0	0	0	4	6
Badgers (*n* = 5)	*Ancylostomatidae* (5) *Capillaria* (1) Coccidia (1) *Strongyloides* (1)	*Crenosoma* (4)	2	0	0	1	0	1	0
Porcupines (*n* = 4)	Negative	Negative	1	1	0	ND ^	ND	ND	ND
Marten (*n* = 1)	Negative	Negative	1	0	0	ND	ND	ND	ND

## Data Availability

Not applicable.

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
