# Peer review of "A One-Year Retrospective Analysis of Viral and Parasitological Agents in Wildlife Animals Admitted to a First Aid Hospital"

_animals, 2023, doi:10.3390/ani13050931_

Round 1
Reviewer 1 Report
The surveillance of viral and parasitic agents in wildlife worldwide is very important, since wild animals can be considered as natural reservoir hosts of several pathogens and as effective vectors of zoonotic diseases that pose an important risk for domestic animals and humans as well. This article describes prevalence of several viral and parasitic agents in rescued Italian wild animals. However, I do suggest certain things, which need attention, improvement and clarification to support and strengthen the overall impact of the article.
Points for attention:
Title: Authors should think about changing the title.
Simple summary:
It would be good to add that nowadays, wild animals are entering the outskirts of cities in search for food and possible contacts with people are not excluded.
Abstract:
Lines 34-36: It would be good if the authors would state that besides the importance of zoonotic pathogens detection in tested wildlife species this study represents the first molecular identification of G. doudenalis sub-assemblage AI and D. capreolus in roe deer in Italy.
Line 36: I would remove the following sentence from the Abstract “All viral and parasitological sequences were deposited in the GenBank database.”
Keywords: To increase article’s searchability it would be good to add more keywords, for example: interspecies transmission, zoonotic potential
Introduction:
Line 44: Remove point after worldwide
Line 62: It would be good if the authors would add that wild animals are entering the outskirts of cities in search for food.
Line 77: Remove one “,” after addition
It would be good if the authors would describe analyzed viral and parasitic agents (some parts from discussion should be removed to the introduction)
Materials and Methods:
Line 125: remove “,” after reference 18
Results:
Lines 282-284: Remove to the chapter M&M (2.2. Enrolled animals)
Table 5 can be summarized in few sentences or removed to the supplementary data.
Discussion:
Line 401: Serological surveys in wildlife are diagnostic tool.
Lines 446-450: Move to the introduction.
Lines 646-470: Move to introduction.
Lines 493-497: Move to the introduction.
Lines 550-555: Move to the introduction.
Also, description of parasite agents should be removed to introduction.
Discussion is too long and confusing. Reorganization needed.
Conclusions:
/
Conclusive remarks: Detection of specific antibodies isn’t virological investigation (M&M, Results). Maybe I missed it, but the Table 3 is not mentioned in the text. Table 7 (line 335) is mentioned in the text before Table 6 (line 366). Maybe authors could switch table numbers. The article needs several improvements in the English Style, as well as formatting to improve the readability of the document.
Author Response
Title: Authors should think about changing the title:
- The new title is “A one-year retrospective analysis of viral and parasitological agents in wildlife animals admitted to a first aid hospital”.
Simple summary:
It would be good to add that nowadays, wild animals are entering the outskirts of cities in search for food and possible contacts with people are not excluded.
- Done
Abstract:
Lines 34-36: It would be good if the authors would state that besides the importance of zoonotic pathogens detection in tested wildlife species this study represents the first molecular identification of G. doudenalis sub-assemblage AI and D. capreolus in roe deer in Italy.:
- The information on the first identification of G duodenalis is already included in the abstract (line 36-37)
Line 36: I would remove the following sentence from the Abstract “All viral and parasitological sequences were deposited in the GenBank database.”
- Done (please see line: 37-38)
Keywords: To increase article’s searchability it would be good to add more keywords, for example: interspecies transmission, zoonotic potential:
- Done (please see line: 40-41)
Introduction:
Line 44: Remove point after worldwide
- Done
Line 62: It would be good if the authors would add that wild animals are entering the outskirts of cities in search for food.
- Done (please see line: 64-65)
Line 77: Remove one “,” after addition
- Done
It would be good if the authors would describe analyzed viral and parasitic agents (some parts from discussion should be removed to the introduction)
- Done (please see line: 74-168)
Materials and Methods:
Line 125: remove “,” after reference 18
- Done
Results:
Lines 282-284: Remove to the chapter M&M (2.2. Enrolled animals)
- Done (please see line: 243-245)
Table 5 can be summarized in few sentences or removed to the supplementary data.
- We moved table 5 to supplementary data
Discussion:
Line 401: Serological surveys in wildlife are diagnostic tool.
- Done
Lines 446-450: Move to the introduction.
Lines 646-470: Move to introduction.
Lines 493-497: Move to the introduction.
Lines 550-555: Move to the introduction.
Also, description of parasite agents should be removed to introduction.
- Done (please see line: 74-168)
Discussion is too long and confusing. Reorganization needed.
- Done
Conclusions:
Conclusive remarks: Detection of specific antibodies isn’t virological investigation (M&M, Results). Maybe I missed it, but the Table 3 is not mentioned in the text. Table 7 (line 335) is mentioned in the text before Table 6 (line 366). Maybe authors could switch table numbers. The article needs several improvements in the English Style, as well as formatting to improve the readability of the document.
- The name of “virological investigation” sections have been renamed. Table 3 has been mentioned in the text. Table 6 and 7 is the same, so the text has been changed. The English style has been improved.

Reviewer 2 Report
It is always nice to see information on pathogenic agents in wildlife, as this is a major area which is largely un-investigated, and the study by Pacini et al., is a great study to fill that gap.
The authors should be commended on their study, and the huge number of pathogens which they have investigated and the manuscript which they have produced.
I only have a few comments and they are all minor, largely around typos and grammatical suggestions where I have suggested an alteration for the authors
Line 12- space needed between ‘expand’ and ‘and’
Line 18- animals is a typo
In the abstract, many of the Latin names need to be put into italics
Line 44- full stop before the reference needs deleting
Line 51- can delete ‘yet’ here
Line 53- become will sound better than becomes
Lone 69- posing is a typo
Line 77- two commas after addition, please delete one
Line 88- where most of the inhabited …. (reword)
Line 92- maybe and where untouched areas are very limited? Or few areas uninhabited by humans remain?
Line 95- two commas after size- please delete one
Line 96- think this should be some and not ome?
Line 125- comma and a full stop after [18]- please delete the comma
Line 127- space needed between basis and of
Line 138- Eppendorf tubes here would sound better
Line 163 – two full stops after instructions
Line 169- space needed between of and data
Line 182- a space is needed between Germany and secondary
Is it also possible to include the reagents for the PCR and cycling conditions in here (unless I missed them)
Line 220- humane euthanasia due to severe … (reword)
Line 230- stored for 24-48 hours (reword)
Line 234- Latin names need to be in italics
Line 241- can the reagents be included here too please
Line 251- how did you quality check the sequences?
Line 268 and other places, could you please replace “ with seconds?
Line 282- enrolled were hospitalised because they were injured (reword)
Line 291- rescued in zone B (reword)
Table 5- length is a typo
Line 340- 2, 1 and 3 parasites seems an odd order?
Line 367- maybe showed positive may sound better than resulted?
Line 380- Latin names need to be in italics
Line 401- space needed between wildlife and are
Line 502- yet can be deleted
Line 520- with a prevalence in line ….. (reword)
Line 529- space needed between only and been
Line 537- carnivores
Line 552- differential is a typo
Line 562- space needed between study and All
Line 563- together is a typo
Line 567- can be controlled in domestic dogs … (Reword)
Line 571- space needed between been and detected
Line 580- Vulpes vulpes needs to be in italics
Line 584- with a direct life cycle…. (reword)
Line 608- has a direct life cycle (reword)
Line 611- as a parasitic agent (reword)
Line 647- usually with a high prevalence … (reword)
Line 659- have an indirect life cycle (reword)
Line 660- has a direct life cycle ..(Reword)
Line 664- did not allow differentiation from one species…. (reword)
Line 692- patterns
Line 701- as an in clinic … (Reword)
Line 702- has also been successfully used in other host …(reword)
Line 707- allowed detection of two additional…. (reword)
Line 718- with a direct orofaecal …(Reword)
Line 729- can delete also
Line 739- Although is a typo
Line 748- focal points for collecting …. (reword)
Author Response
It is always nice to see information on pathogenic agents in wildlife, as this is a major area which is largely un-investigated, and the study by Pacini et al., is a great study to fill that gap.
The authors should be commended on their study, and the huge number of pathogens which they have investigated and the manuscript which they have produced.
I only have a few comments and they are all minor, largely around typos and grammatical suggestions where I have suggested an alteration for the authors.
Line 12- space needed between ‘expand’ and ‘and’
- Done
Line 18- animals is a typo
- Done
In the abstract, many of the Latin names need to be put into italics
- Done
Line 44- full stop before the reference needs deleting
- Done
Line 51- can delete ‘yet’ here
- Done
Line 53- become will sound better than becomes
- Done
Lone 69- posing is a typo
- Done
Line 77- two commas after addition, please delete one
- Done
Line 88- where most of the inhabited …. (reword)
- Done
Line 92- maybe and where untouched areas are very limited? Or few areas uninhabited by humans remain?
- The sentence has been reworded
Line 95- two commas after size- please delete one
- Done
Line 96- think this should be some and not ome?
- Done
Line 125- comma and a full stop after [18]- please delete the comma
- Done
Line 127- space needed between basis and of
- Done
Line 138- Eppendorf tubes here would sound better
- Done
Line 163 – two full stops after instructions
- Done
Line 169- space needed between of and data
- Done
Line 182- a space is needed between Germany and secondary
- Done
Is it also possible to include the reagents for the PCR and cycling conditions in here (unless I missed them)
- The melting temperatures were reported in table S2. The sentence “PCR were performed according to the manufacturer instruction of kits” has been added.
Line 220- humane euthanasia due to severe … (reword)
- The sentence has been reworded
Line 230- stored for 24-48 hours (reword)
- Done
Line 234- Latin names need to be in italics
- Done
Line 241- can the reagents be included here too please
- To carry out the procedure, we strictly followed the instructions of the kit, as explicitly stated in the text.
Line 251- how did you quality check the sequences?
- We have explained in the text (please see line: 378-380)
Line 268 and other places, could you please replace “ with seconds?
- Done
Line 282- enrolled were hospitalised because they were injured (reword)
- Done
Line 291- rescued in zone B (reword)
- Done
Table 5- length is a typo
- Done
Line 340- 2, 1 and 3 parasites seems an odd order?
- It’s correct
Line 367- maybe showed positive may sound better than resulted?
- Done
Line 380- Latin names need to be in italics
- Done
Line 401- space needed between wildlife and are
- Done
Line 502- yet can be deleted
- Done
Line 520- with a prevalence in line ….. (reword)
- Done
Line 529- space needed between only and been
- Done
Line 537- carnivores
- Done
Line 552- differential is a typo
- Done
Line 562- space needed between study and All
- Done
Line 563- together is a typo
- Done
Line 567- can be controlled in domestic dogs … (Reword)
- Done
Line 571- space needed between been and detected
- Done
Line 580- Vulpes vulpes needs to be in italics
- Done
Line 584- with a direct life cycle…. (reword)
- Done
Line 608- has a direct life cycle (reword)
- Done
Line 611- as a parasitic agent (reword)
- Done
Line 647- usually with a high prevalence … (reword)
- Done
Line 659- have an indirect life cycle (reword)
- Done
Line 660- has a direct life cycle ..(Reword)
- Done
Line 664- did not allow differentiation from one species…. (reword)
- Done
Line 692- patterns
- Done
Line 701- as an in clinic … (Reword)
- Done
Line 702- has also been successfully used in other host …(reword)
- Done
Line 707- allowed detection of two additional…. (reword)
- Done
Line 718- with a direct orofaecal …(Reword)
- Done
Line 729- can delete also
- Done
Line 739- Although is a typo
- Done
Line 748- focal points for collecting …. (reword)
- Done
